# Deciphering Seasonal Patterns in Animal Feeding: A Mechanistic Approach to Analyzing the Restricted Growth of Iberian Pigs

**DOI:** 10.3390/ani14233431

**Published:** 2024-11-27

**Authors:** Fernando Sánchez-Esquiliche, Chelo Ferreira, Rosa Nieto, Luisa Ramírez, Gema Matos, Ana Muniesa

**Affiliations:** 1Sánchez Romero Carvajal, SRC, Jabugo, 21290 Huelva, Spain; 2Department of Animal Pathology, Facultad de Veterinaria, Universidad de Zaragoza, 50013 Zaragoza, Spain; animuni@unizar.es; 3Department of Applied Mathematics, Facultad de Ciencias, Universidad de Zaragoza, 50009 Zaragoza, Spain; 4Department of Nutrition and Sustainable Animal Production, Estación Experimental del Zaidín, Consejo Superior de Investigaciones Científicas, CSIC, Profesor Albareda, 1, 18008 Granada, Spain; rosa.nieto@eez.csic.es

**Keywords:** Iberian pigs, mathematical modeling, compartmental models, growth prediction, protein deposition, nutritional management, age–weight relationship

## Abstract

Animal growth is a critical area of research for both scientists and livestock producers, and is influenced by genetic, nutritional, and environmental factors. However, there is a notable lack of studies on the association between growth within animal production systems and local environments. The available literature on native breeds is limited, primarily due to the reliance of autochthonous breeds on local resources. A prime example is the acorn-fed Iberian pig, whose growth phases prior to the finishing period, in which they consume acorns, remain inadequately explored. This study proposes the application of non-linear mathematical models to elucidate this growth process. Additionally, it adopts a mechanistic perspective, conceptualizing the total growth of the animal as the aggregate of the growth of its various body components. By advancing our understanding of these dynamics, we aim to contribute valuable insights to optimize the management practices used in Iberian pig production systems.

## 1. Introduction

For centuries, animal scientists have sought to uncover the complexities of animal growth, where variations within a species are largely influenced by genetics and nutrition. Mathematical models have been used to uncover the underlying processes involved in animal production [1], particularly in predicting nutrient requirements [2], feed intake, and growth [3]. Several mathematical models have been used to compare pig growth, involving factors such as genetics, sex, and growth stage. Some models consider the pig’s body as a unit (non-compartmental models), while others divide it into compartments based on its chemical composition (compartmental models). The latter provide more detailed insights into nutrient utilization, although they require breed-specific knowledge of genetic potential and nutritional needs [4].

In terms of animals’ relationships with humans, there are two extremes: wild animals capable of adapting to seasonal changes and the variable availability of food; and laboratory animals living under controlled conditions. Local breeds, such as the acorn-fed Iberian pig, are somewhere in between. These environmentally adapted breeds have received little scientific attention [5].

In traditional Iberian pig production, there are important changes in growth (feeding) management between 23 kg and 100 kg of body weight (BW) [6]. Between these two BWs, pigs are fed in a restricted manner. The growth phase of free-range Iberian pigs remains poorly studied, probably due to its variability [7]. Following this phase of restricted growth, and coinciding with the season of acorn availability, the traditional open-air free-range (*montanera*) finishing phase takes place, generally between October and March.

Protein deposition is crucial in order for an animal to achieve a larger overall size, since the organism prioritizes muscle over fat tissue growth [8]. Moreover, the Iberian pig reaches its maximum protein deposition capacity at a relatively smaller size compared to commercial pig breeds [9]. The traditional acorn diet provided during the period to subsequent feed restriction, once the animal has achieved its maximum protein deposition capacity, is characterized by a deficiency in protein and lysine [10] and underlines the importance of adequate protein intake prior to the pre-finishing phase [6]. This strategic approach aims to ensure that the ideal slaughter weight can be achieved according to market objectives.

Although mathematical models have been applied to a variety of pig growth scenarios, there is a lack of studies on local pigs reared under season-dependent feeding conditions. This research aims to shed light on the traditional outdoor growth phase prior to the open-air free-range finishing period in Iberian pigs by evaluating different mathematical models based on BW and body component distribution.

## 2. Materials and Methods

### 2.1. Animal Data

This study was carried out in a commercial population of Iberian pigs (SRC line) [11]. Sixteen weight–age databases (published and non-published) of pigs registered between 2012 and 2022 were used. A total of 5329 pigs, individually identified at birth with at least two ear tags, were weighed between 2 and 9 times during their lifetime, from 73 days to carcass weight at slaughter.

The pigs were reared on seven commercial farms located in southwestern Spain. The complete database contains 19,185 weight–age records, with 2901 commercial carcass weights at slaughter. All animals were slaughtered in the same commercial slaughterhouse.

### 2.2. Productive Phase Categorisation

The first categorization of data was based on the productive phase. For this article, the growing and pre-finishing phases were studied from approximately two months post-weaning until before the traditional finishing phase where the pigs are fed with acorns in the Mediterranean forest.

During the pre-finishing phase, pigs are reared outdoors, with access to pasture (Figure 1) and restricted dietary supplements [12]. The end of this phase and, consequently, the beginning of the next phase of acorn-fed finishing coincides with the availability of acorns, from October to December. At the end of the studied period, the pigs in this database were aged between 304 and 590 days and weighed 106.37 ± 14.53 kg. This variability is due to the seasonality of acorn availability (October to March) and to legislative constraints on the acceptable weight range and minimum individual age of pigs [13], together with the low reproductive seasonality of this particular pig breed [14].

### 2.3. Diet and Nutrition

During the growing and pre-finishing phases, the pigs had access to pasture and they were provided with a balanced diet that they were restrictedly fed as described previously [15]. The dietary formulas used fluctuated in terms of their ingredients (barley, wheat, soybean meal, maize, etc.) and the inclusion percentages of these components, although the final nutrient composition remained stable (Table 1).

### 2.4. Data Categorisation and Allometric Analysis

To better analyze the data, the databases were categorized into three age groups according to the pigs’ ages at the beginning of the finishing period, following Daza et al. [16]. However, unlike the aforementioned studies, the datasets used in this study also include information on older pigs at the end of pre-finishing period, resulting in the definition of two additional groups (Table 2). In the oldest group, there was a lack of BW data prior to 187 days of age.

Several empirical equations previously described by Nieto [9] were applied (Table 3). These models’ predictions were performed using animals from the same genetic population as those of the present paper. With these algorithms, the body components, including the cold carcass weight with skin and hair and without feet and head [17], and the chemical components of the carcass, including fat, protein, water, and ash, were estimated based on empty BW [18].

### 2.5. Mathematical Modeling and Statistical Analysis

To analyze the weight data obtained during the restricted growth of Iberian pigs, several previously published mathematical models were applied [4,19,20,21,22] whilst also taking into account the influence of seasonal food availability. These models offer the flexibility to either pre-fix the asymptote (*A*) or not, resulting in non-compartmental algebraic models, i.e., models that attempt to predict the growth of the pig as a whole. The data were categorized according to the final age at the end of the pre-finishing phase, with different models being constructed for the same phase.

On the other hand, by applying the empirical equations mentioned above, the relative growth of different body chemical components (protein, fat, water and ash) was also estimated and compartmental models have been applied to these data. Accordingly, models based on the growth curve have been developed for BW and the weights of the carcass’s chemical compartments. To estimate the total carcass weight, following a mechanistic approach, the models were derived from the sum of the aforementioned compartmental models.

To assess the performance of the model, we followed the approach recommended by Mayer [23] by regressing the observed values (variable Y) against values predicted by the model (variable X). In addition, we used the root mean square error of the regression (RMSE) as a measure of goodness of fit. In order to compare models with different scales or dimensions, we used a relative index by dividing the RMSE by the asymptotic value: RMSE/*A*. The *lm* and *nls* functions in RStudio (R software environment, version R.4.4.1) were used for model fitting, and *persp3d* and *trans3d* were employed to create surface plots (Appendix A)) and Excel (Microsoft Office, version 2013) was used for the statistical analysis.

For better precision and clarity, Greek letters have been used to name the groups of pigs, Roman numerals to denote the different models, and the prime symbol (′) to indicate models based on the same dataset with different asymptotic weights. *A*, *b*, and *k* are fitted unitless parameters for the different non-linear models used.

## 3. Results

### 3.1. Assessment of the Age-Weight Relationship: Model Selection

To determine the best-fit model, we used the full age and BW datasets to evaluate three growth models: the Gompertz model [I], the Von Bertalanffy model [II], and the Brody model [III] (Table 4). The age and BW data indicated that the best fit would be obtained with the Gompertz model, which showed the lowest RMSE of the three models evaluated (9.97 vs. 10.98 vs. 16.36, respectively).

Gompertz function [4,20]
$$y(t)={Ae}^{{-e}^{b-kt}}$$

Van Bertalanffy function [21]
$$y(t)=A{\left(1-be^{-kt}\right)}^{3}$$

Brody function [19]
$$y(t)=A(1{-be}^{-kt})$$
where *y*(*t*) is weight, *A* is asymptotic weight, and t is the age; *A*, *b*, and *k* are the parameters fitted by the model.

The curve plotted using the Gompertz model exhibits distinct biological features, as detailed in Table 5 which presents several mathematical parameters of the model, including the inflection point where the curve changes its slope, corresponding to a specific age and BW.

Considering the asymptotic BW (A) we applied two fitting models, namely [IV] (with fixed asymptotic weight) and [IV′] (with free asymptotic weight) to the pre-finishing dataset based on the Gompertz curve (Table 4). These fits provided eight additional non-compartmentalized mathematical models of age and BW. Notably, model [IV′] showed the best fit among those evaluated, yielding an RMSE of 6.77.

Table 2 shows the number of animals in each age group at the end of the pre-finishing period, as well as some descriptive performance data and other descriptive values. The fitting, using the Gompertz model, was performed for each age group. The last three columns of Table 2 show the estimated BW at the average age just before the start of the subsequent outdoor phase (open-air, free-range) according to each of the models developed. Applying models [IV] and [IV′] to the average age of each group in Table 2 revealed that the calculated value closely aligns with the value generated by model [IV′]. Furthermore, and according with the information in Table 2, the differences between the observed pre-finishing BW values for each age group (α, β, γ, δ, and ε) with each model ([V], [VI], [VII], [VIII], and [IX]), respectively, were closer to the value predicted by the model for that specific group than with the predictions of either model [IV] or [IV′] (Figure 2).

Sequentially, groups α to ε show a steady increase in age. The same is true for BW, except for group ε which has a lower weight than the δ group. The ε group showed the greatest heterogeneity in the final BW (SD 17.12) but lacks BW data for ages below 187 days (Table 2). For the ε group, two approaches were followed, where either A = 228 or A = 205, which as the maximum BW found in the ε group (models IX and IX′, respectively). This distinction guides model construction, revealing an approximate 20 kg reduction in the expected BW at the end of the pre-finishing phase when applying the [IX′] model (Table 4), which aligns more closely with the observed BW. The models derived for this group show a sub-optimal statistical performance for the full dataset used within this study.

### 3.2. Compartmental Models: Protein Growth with Restricted Growth

The chemical components of the carcass, protein, fat, water, and ash were estimated using the algorithms described by Nieto [9] (Table 3). Models based on the Gompertz curve were constructed for carcass protein ([X], [XI], [XII], [XIII], [XIV], and [XV]) and for fat, water, and ash ([XVI], [XVII], and [XVIII]), respectively (Table 4). Compartmental models focused on protein gain were developed, spanning from [XI] to [XIV] for the groups from α to ε. Model [X] was constructed using the entire database and showed the most favorable performance in terms of RMSE (RMSE = 0.38 and RMSE/A = 3.65%), obviating the need to split the database between different age groups, as required in the non-compartmental model calculations described above (Figure 3). The inflection points for carcass protein and carcass water were quite close (Table 6), in line with the close relationship between the two parameters during muscle growth. In this dataset, the inflection point happens at around 157.5 or 166.9 days of age for protein and water (Table 6). The inflection point for ash deposition, related to bone mineral deposits and body development at early ages, was also close to the previous two (218.6 days of age). However, the inflection point for fat deposition occurred much later (361.1 days of age).

The estimated body protein data for the ε group were discarded as they may be biologically meaningless according to the information obtained in this study regarding the protein deposition inflection point and the lack of BW data for ages below the inflection point.

### 3.3. Ensemble Models

For whole carcass predictions, two tests were carried out to aggregate the chemical components of the carcass. In the first test [XIX], we combined the Gompertz models [X], [XVI], [XVII], and [XVIII]. In the second test [XX], we used the sum of the Gompertz models for protein [X], water [XVII], and ash [XVIII], together with the fat estimate based on BW [9]. The lowest RMSE was obtained for the second model [XX], and the RMSE/A for this model was the best among those calculated in this work (Figure 4 and Figure 5).

Carcass ash: model [XVIII]
$$y(t)=({3.18e}^{{-e}^{0.9372-0.0043t}})$$

Carcass without fat and water: models [XVII] + [XVIII]
$$y(t)={(10.38e}^{{-e}^{0.8003-0.0051t}})+({3.18e}^{{-e}^{0.9372-0.0043t}})$$

Carcass without fat: models [X] + [XVII] + [XVIII]
$$y(t)={(10.38e}^{{-e}^{0.8003-0.0051t}})+({37.91e}^{{-e}^{0.7055-0.0042t}})+({3.18e}^{{-e}^{0.9372-0.0043t}})$$

Carcass: model [XX]
$$y(t)={(10.38e}^{{-e}^{0.8003-0.0051t}})+(0.0191\times {\left(eBW\right)}^{1.6725})+({37.91e}^{{-e}^{0.7055-0.0042t}})+({3.18e}^{{-e}^{0.9372-0.0043t}})$$

Four views of the carcass: model [XX]
$$y(t)={(10.38e}^{{-e}^{0.8003-0.0051t}})+(0.0191\times {\left(eBW\right)}^{1.6725})+({37.91e}^{{-e}^{0.7055-0.0042t}})+({3.18e}^{{-e}^{0.9372-0.0043t}})$$

## 4. Discussion

Mathematical vs. Biological Interpretation.

The Gompertz model [4] was selected because of its reduced RMSE compared to the Van Bertalanffy [21] or Brody [19] curves when applied to our data. Other authors, such as Tedeschi [24] in cattle, or Usero-Alonso et al. [25] in crossbred Iberian pigs, also chose this model. To calculate the total BW models, *A* = 226 (asymptotic or maximum weight) was used. This value, although somewhat higher, is in line with the findings of Vouri et al. [26] and Casas et al. [4] for conventional pigs, and Usero-Alonso et al. [25] for crossbred Iberian pigs, who reported values close to 220, 222.7, and 218 kg BW, respectively.

To assess the above data and find the best model to explain the BW–age relationship, two models were tested. In the first [IV] the asymptote *A* = 226 was fixed. The second model [IV′] was calculated without setting any parameters and the *A* estimation obtained was 156.1. This may not make sense mathematically or even physiologically. However, when applied to the average age at the end of the pre-finishing phase (Table 2), model [IV′] gave BW results closest to the average of the dataset. This suggests that, in farming practice in Iberian pig systems, such as open-air free-range system, where acorn availability is exploited for fattening and finishing pigs, farmers must consider factors beyond the growth rate, including anticipated feed availability, and modulate animal growth accordingly; in other words, part of the pig growth/fattening process is modulated by farmers in order to take advantage of the subsequent availability of natural feed. In a meta-analysis, Sánchez-Esquiliche et al. [7] found that the final BW before finishing fluctuated between 103 and 117 kg for ages between 283 and 418 days, respectively, based on the weights of 3659 pigs. In contrast, Daza et al. [27], in a single study involving 24 pigs, reported mean weights of 99.7 and 102.6 kg for a similar age range, which align more closely with the findings of this study. The consistency in the final pre-finishing BW over the consecutive years covered by our research, albeit using pigs from the same genetic line, probably contributed to the greater uniformity seen compared to the dataset used in the aforementioned meta-analysis. Segmenting the growth curve into different age intervals increased the accuracy of its fit for the predicted BW of pigs under restricted feeding. This approach carefully considers the age at which each animal completes its growth stage. A discrete approach, adopted by Sánchez-Esquiliche et al. [7] and Usero-Alonso et al. [25] in studies on Iberian and crossbreed Iberian pig, respectively, involves segmenting growth into several phases, and responding to changes in management, nutritional changes, and feed restriction. Given the specific management practices applied during growth, dietary restriction in these animals requires the development of multiple models, as a single curve is not sufficient to comprehensively explain their growth dynamics. Thus, model [V] and model [IX] deviate significantly from model [IV′].

Model [V] was calculated using the weight of the youngest group of animals at the end of the pre-finishing phase in this study (α group), while model [IX] corresponds to the oldest group ε. The α group was deliberately subjected to accelerated growth with a slight food restriction. Born in the autumn, these pigs were fed mature acorns during the outdoor finishing phase of the year following their birth, which started at around one year of age. In contrast, the ε group and model [IX] comprise pigs born in the spring. This group could not be finished with the acorn crop from the year of their birth, but rather with the following year’s crop, therefore the finishing phase started at 19 months, resulting in prolonged growth periods and deliberately slower growth rates than those observed in the other groups.

Categorizing the database according to the age of the individuals at the end of the study period improved our biological understanding of the models, although this is not necessary when using compartmental models, like, for example, in the protein deposition model. This disparity is due to the fact that the overall weight gain may not lead to increased protein deposition, but could instead reflect early fattening. García-Contreras et al. [15] observed similar findings in a study involving restrictedly fed pigs from the same population as that studied here, in which they tried to develop predictive algorithms for body composition. The observed that protein gain was related to increases in height or stature, but not to increases in back fat or total BW. This premature fattening may negatively influence subsequent development. In this regard, Sánchez-Esquiliche et al. [7] recorded a suboptimal performance in pigs that started finishing at a younger age, which could be due to the above.

On the other hand, model [IX] deviates significantly from the rest of the models. This difference is evident from the initial BW–age data. In the absence of BW data from animals younger than 187 days for this group, it is not appropriate to compare this model with the others (Figure 2 and Figure 3), especially the compartmental models of protein growth at young ages.

Early protein restriction could negatively affect the subsequent growth of the animals [17,18]. Wild animals often face periods of low food availability due to seasonal weather patterns. In times of adversity, wild animals tend to prioritize vital functions over those less critical to immediate survival. Fattening or reserve accumulation, considered a secondary function, is of less biological or physiological importance than lean tissue growth. In this vein, growth could also be secondary in times of severe food shortage.

Sánchez-Esquiliche et al. [7] provided data for animals at 69.1 days of age and a BW of 21.1 kg. When we applied this age to our models, we obtained comparable data: 20.1, 20.8, 22.5, 20.9, and 22.2 kg of BW for this age in models [IV′], [V], [VI], [VII], and [VIII], respectively. As mentioned above, it is important to note that model [IX] is not applicable for estimating the BW for at this age, as it was constructed with data collected from pigs 187 days and older with no previous BW data recorded.

Feed restriction is not a common practice in commercial pig farming. In traditional livestock production, closely linked to environmental conditions and seasonal harvests, there are examples when nutrient availability is insufficient for optimal feeding. Several critical developmental stages, such as the formation of muscle structure, can be affected by this nutrient limitation. Traditional breeds, such as the Iberian pig, which are not selected for their enhanced protein deposition, also tend to exhibit slower growth trends accompanied by a tendency towards fattening under conditions of food scarcity [5].

Local breeds exhibit low protein deposition capacities [5]. In this study, it was also observed that the inflection point for protein deposition occurred at 157.5 days of age, and the inflection point for water occurred at 166.9 days of age. It is important to note that these animals are typically slaughtered between 420 and 600 days of age [6], meaning they spend most of their productive life after this peak in a protein deposition phase. Previous studies suggested a high energy cost associated with muscle protein turnover in with Iberian pigs [28], which might contribute to the higher production costs for this native breed.

Fat deposition, on the other hand, seems to be less dependent on age and more influenced by excess resource intake once the animal’s basic needs are met. This factor is crucial for differentiating the meat products derived from these animals, either due to the unsaturated fat profile or the intramuscular fat distribution achieved by the end of the production process [29].

### Ensemble Models

Models [XIX] and [XX] are a combination of several other models. To achieve more accurate predictions of carcass and, consequently, BW, it is necessary to combine age-based and weight-based models. This approach is facilitated by accurate in vivo estimates of the body components of the Iberian pig.

To construct model [XX], it was necessary to integrate non-linear models for estimating protein, water, and ash content based on age, together with an estimation of the fat content based on BW (Figure 4). This resulted in a three-dimensional model, with age on the *X*-axis, BW on the *Y*-axis, and carcass weight, as predicted by model [XX], on the *Z*-axis (Figure 5). By using these models, we can predict the structure of the animal (based on protein, water, and ash) and therefore anticipate the growth or fattening it may experience at later stages, taking into account the initial fat and available feed.

## 5. Conclusions

A comprehensive estimate of Iberian pig BW can be achieved by employing five categorical models classified according to the age at which pigs complete their pre-finishing stages. To understand weight changes in Iberian pigs with a restricted diet in the pre-finishing period (prior to open-air free-range finishing), an assembly of compartmental models predicting protein, water, and ash growth in relation to age is needed, using Gompertz-based models, while for adipose tissue growth, BW-based predictions were found to improve the model.

When Iberian pigs are subjected to severe feed restriction before 156 days of age, protein deposition may be compromised, either due to a lack of essential amino acids, or a lack of energy, or both. This restriction could negatively affect these pigs as their potential for protein growth cannot be optimized at this age and they may not end up reaching their target BW at later stages. Further research with older pigs is needed to better understand their growth patterns. The assembled models can potentially be extrapolated to other traditional pig breeds, facilitating an understanding of growth patterns under conditions of natural resource use. In addition, there is a need to move towards a mechanistic model to increase our understanding of the underlying physiological mechanisms governing the growth of these pigs.

## Figures and Tables

**Figure 1 animals-14-03431-f001:**
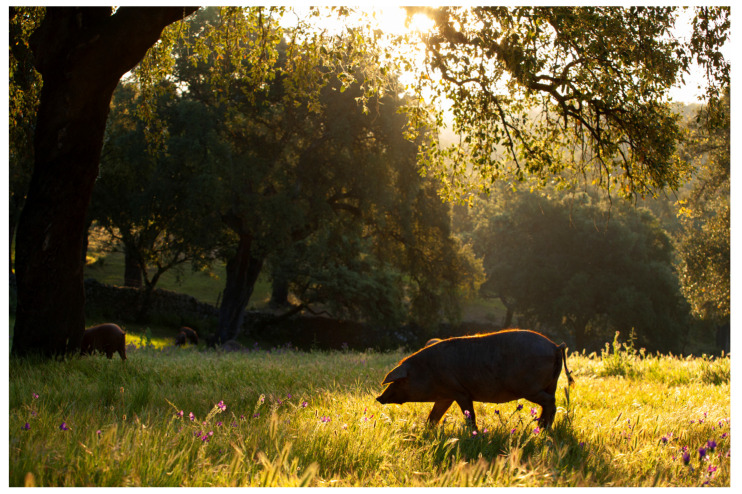
SRC Iberian pig grazing during the spring, in its growing stage. Photo by Kris Ubach, owned by Sánchez Romero Carvajal-Jabugo enterprise.

**Figure 2 animals-14-03431-f002:**
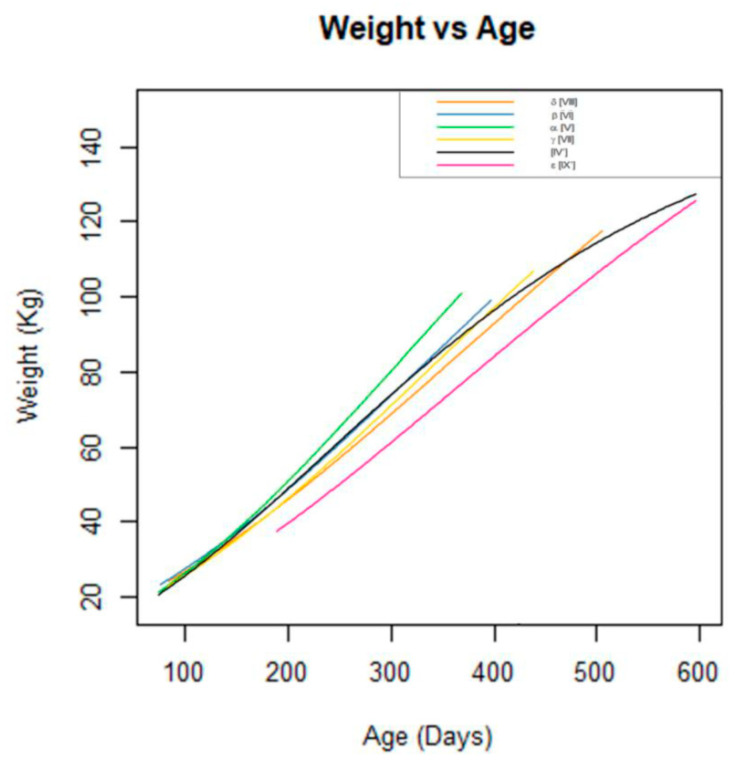
Gompertz BW models in the restricted growth of Iberian pig. α, β, γ, δ, and ε final pre-finishing age groups. [IV′] Non-compartmental Gompertz pre-finishing model. [V, VI, VII, VIII, and IX′] Non-compartmental Gompertz models for the groups α, β, γ, δ, and ε, respectively.

**Figure 3 animals-14-03431-f003:**
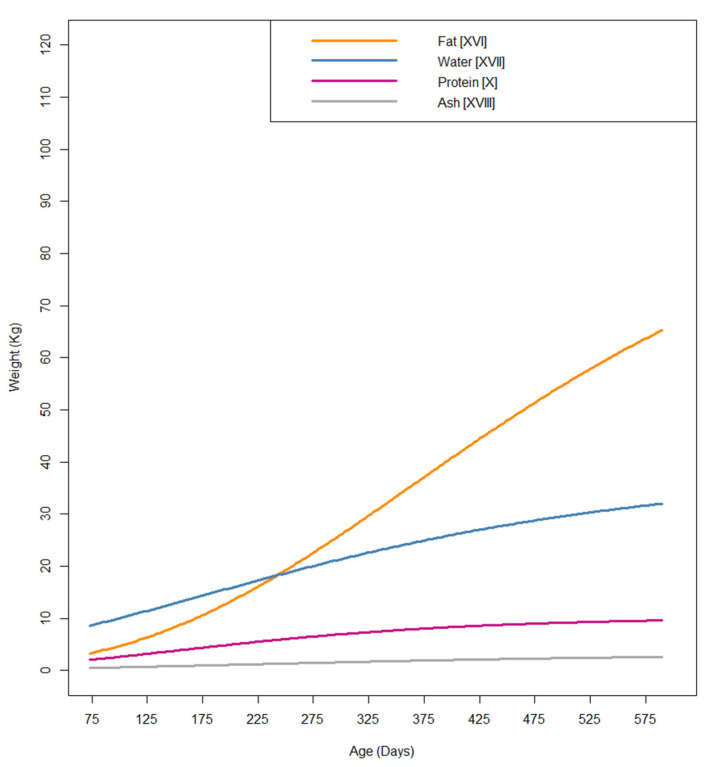
Gompertz carcass components models in the growth of Iberian pig.

**Figure 4 animals-14-03431-f004:**
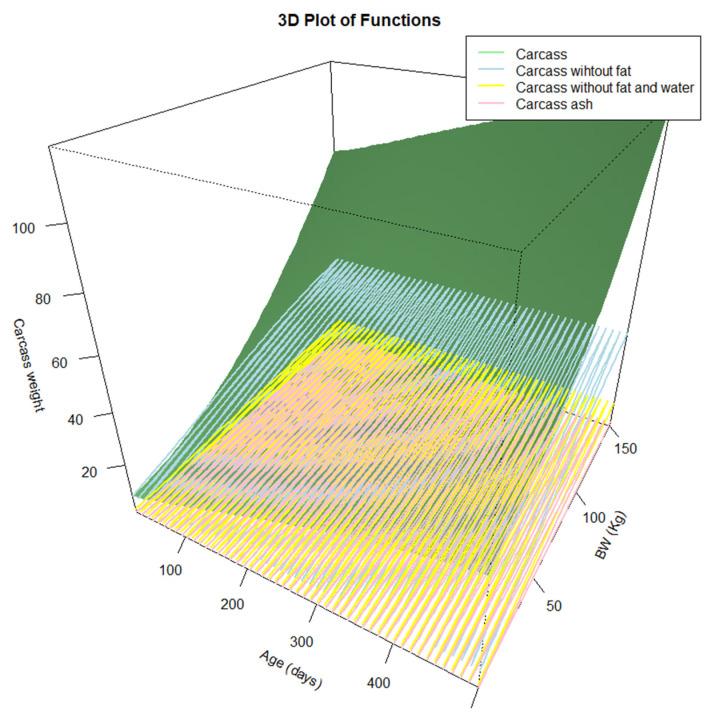
Ensemble model: Mechanistic model to explain the carcass growth of Iberian pigs under food restriction.

**Figure 5 animals-14-03431-f005:**
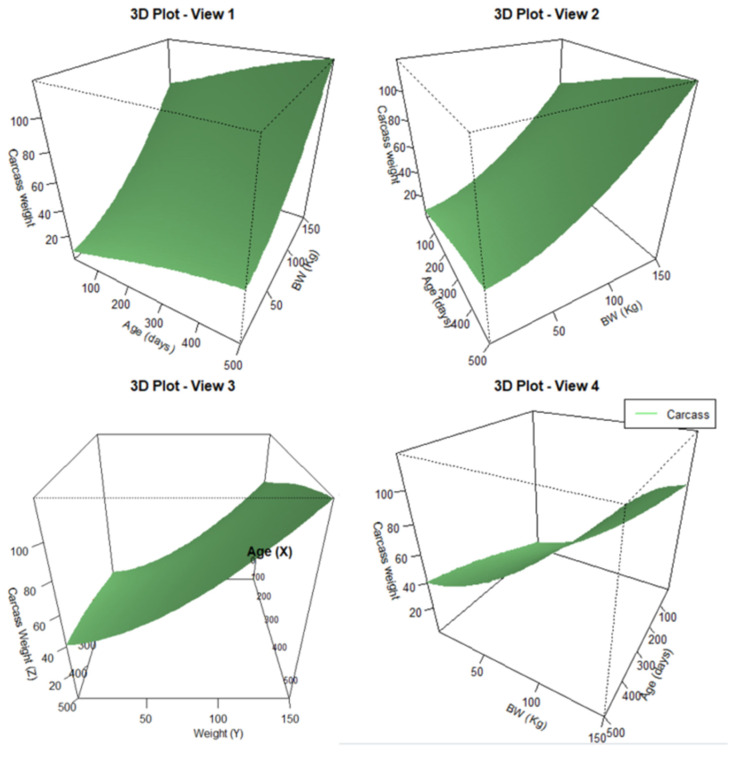
Mechanistic model to explain the carcass growth of Iberian pigs under food restriction.

**Table 1 animals-14-03431-t001:** Nutritional composition of the feeds used for pigs in the databases used in the present work within the body weight (BW) intervals of 25–60 kg and over 60 kg until the open-air free-range phase.

Nutrients	Unit	25–60 kg BW	+60 kg BW
Crude protein	g/kg	130	110
Lysine	g/kg	8.4	7
Methionine	g/kg	2.6	2.3
Methionine + Cisteine	g/kg	5.4	4.5
Threonine	g/kg	5.5	4.6
Triptophane	g/kg	1.6	1.35
Lipids	g/kg	25	45
Crude fiber	g/kg	40	45
Ash	g/kg	46	50
Metabolizable energy	MJ/kg	13.0	13.1

**Table 2 animals-14-03431-t002:** Convergence between the average body weight (BW) attained by Iberian pigs at the pre-finishing growing stage and the BW predicted by different growth models.

Groups	Number of Pigs	Number of Weights	Minimum Age (Days)	Final Pre-Finishing Age Range (Days)	Average Final Pre-Finishing BW (kg, SD)	Predicted BW Model [IV]	Predicted BW Model [IV′]	Predicted BW Models [ ]
α	791	2731	75	[304; 365]	98.73 (11.92)	83.44	89.82	96.92 [V]
β	1379	4848	73	[366; 393]	103.59 (13.05)	90.27	95.80	95.87 [VI]
γ	1351	5047	82	[394; 434]	107.53 (13.96)	97.70	103.89	101.49 [VII]
δ	1388	5183	83	[435; 500]	112.88 (14.24)	108.53	115.19	108.35 [VII]
ε	420	1376	187	[501; 590]	109.59 (17.12)	124.02	127.30	136 [IX]
								115.06 [IX′]
Total	5329	19,185						

α, β, γ, δ, and ε final pre-finishing age groups. [IV, IV′] Gompertz models for all pigs. [V, VI, VII, VIII, IX, IX′] Gompertz models for each group.

**Table 3 animals-14-03431-t003:** Models used for estimating carcass chemical components [9].

Y = aX^b^	X = BWe	
Y	a	b
Cold carcass	0.4776	1.1085
Protein	0.1931	0.8116
Fat	0.0191	1.6725
Water	0.8984	0.7357
Ash	0.0293	0.9226

BWe: empty body weight; BWe = 0.934 × BW^1.007^ [9].

**Table 4 animals-14-03431-t004:** Mathematical models obtained for pre-finishing open-air free-range Iberian pigs: non-compartmental and compartmental (protein, fat, water, and ash) models for all dataset or different final age groups.

	Literature Models	Group	Age Range	Model	A	b	k	RMSE	RMSE/A × 100
Non-compartmental models	Gompertz [4]	All	[73; 682]	[I]	228.0	1.2316	0.0040	9.97	4.37%
	Van Bertalanffy [21]	All	[73; 682]	[II]	228.0	0.7962	0.0034	10.98	4.81%
	Brody [19]	All	[73; 682]	[III]	228.0	1.2987	0.0025	16.36	7.18%
	Gompertz [4]	All	[73; 590]	[IV]	228.0	0.9833	0.0029	7.50	3.29%
		All	[73; 590]	[IV′]	156.1	1.0240	0.0044	6.77	4.33%
		α	[75; 365]	[V]	228.0	1.1258	0.0036	7.15	3.14%
		β	[73; 393]	[VI]	228.0	1.0572	0.0032	7.36	3.23%
		γ	[82; 434]	[VII]	228.0	1.0877	0.0031	7.59	3.33%
		δ	[83; 500]	[VIII]	228.0	1.0458	0.0029	6.64	2.91%
		ε	[187; 590]	[IX]	228.0	1.1269	0.0029	9.70	4.26%
		ε	[187; 590]	[IX′]	205.0	1.1027	0.0031	9.53	4.65%
Protein models		All	[73; 590]	[X]	10.38	0.8003	0.0051	0.38	3.65%
		α	[75; 365]	[XI]	10.23	0.9397	0.0063	0.60	5.91%
		β	[73; 393]	[XII]	10.23	0.8110	0.0053	0.43	4.19%
		γ	[82; 434]	[XIII]	10.23	0.8978	0.0055	0.59	5.74%
		δ	[83; 500]	[XIV]	10.23	0.8567	0.0053	0.54	5.25%
		ε	[187; 590]	[XV]	10.23	1.5412	0.0065	0.99	9.67%
Fat model		All	[73; 590]	[XVI]	96.01	1.5246	0.0042	1.37	1.43%
Water model		All	[73; 590]	[XVII]	37.91	0.7055	0.0042	0.71	1.86%
Ash model		All	[73; 590]	[XVIII]	3.18	0.9372	0.0043	0.22	6.78%
Carcass models	[X] + [XVI] + [XVII] + [XVIII]	All	[73; 590]	[XIX]				1.34	0.91%
	[X]+ fat ^1^ + [XVII] + [XVIII]	All	[73; 590]	[XX]				0.55	0.37%

^1^ Fat model of Nieto [9]: Fat = 0.0191 × (empty BW^1.6725^). *A*, *b*, and *k* are fitted unitless parameters. α, β, γ, δ, and ε final pre-finishing age groups. RMSE: root mean square error of the regression. RMSE/*A*: RMSE divided by the asymptotic value (*A*). [I, II, and III] non-compartmental Gompertz, Van Bertalanffy, and Brody models, respectively. [IV and IV′] non-compartmental Gompertz pre-finishing models. [V, VI, VII, VIII, IX, and IX′] non-compartmental Gompertz models for the groups α, β, γ, δ, and ε, respectively. [X, XVI, XVII, and XVIII] compartmental Gompertz models for de protein, fat, water, and ash, respectively. [XI, XII, XIII, XIV, and XV] compartmental Gompertz protein models for the groups α, β, γ, δ, and ε, respectively. [XIX, XX] Carcass weight models resulting from the assembly of compartmental models for protein, fat, water, and ash.

**Table 5 animals-14-03431-t005:** Mathematical properties of the Gompertz model of animal growth.

MATHEMATICAL PROPERTIES	
ASYMPTOTE	Y = *A*
INFLECTION	t = *b*/*k*; Y = *A*/e
GROWTH RATE	dY/dt = *k* × Y × e(*b* − (*k* × t)) = *k* × Y × ln(*A*/Y)
MAXIMUM GROWTH RATE	(*k* × *A*)/exp
RELATIVE RATE AS A FUNCTION OF TIME (T)	1/Y dY/dt = *k* × e(*b* − (*k* × t))
RELATIVE RATE AS A FUNCTION OF MASS/SIZE (Y)	1/Y dY/dt = *k*(ln*A* − lnY)
AGE AT INFLECTION POINT	*b*/*k*
WEIGHT AT INFLECTION POINT	*A*/e

**Table 6 animals-14-03431-t006:** Parameterization of the Gompertz models.

	Group	Model	Age at Inflection Point (days)	Weight at Inflection Point (kg)	dY/dt Max (kg/day)	50% Maturity	98% Maturity
Non-compartmental models	All	[I]	311.6	83.9	0.332	404.3	1298.0
	All	[IV]	353.4	83.9	0.233	486.0	1756.0
	All	[IV′]	231.2	57.4	0.254		
	α	[V]	308.8	83.9	0.306	409.3	1379.0
	β	[VI]	335.2	83.9	0.264	451.6	1572.8
	γ	[VII]	345.8	83.9	0.264	462.3	1586.0
	δ	[VIII]	359.0	83.9	0.244	484.9	1698.5
	ε	[IX′]	398.3	83.9	0.237	527.9	1777.6
Protein models	All	[X]	157.5	39.1	0.021	223.3	857.3
	α	[XI]	150.0	38.3	0.024	208.5	772.9
	β	[XII]	151.8	38.7	0.020	220.4	882.1
	γ	[XIII]	162.7	39.0	0.021	229.0	869.5
	δ	[XIV]	162.5	38.9	0.020	232.0	902.5
Fat model	All	[XVI]	361.1	89.0	0.149	447.9	1285.3
Water model	All	[XVII]	166.9	41.3	0.059	253.5	1089.7
Ash model	All	[XVIII]	218.6	54.3	0.005	304.0	1128.5

α, β, γ, δ and ε final pre-finishing age groups. [I] Non-compartmental Gompertz, model. [IV and IV′] Non-compartmental Gompertz pre-finishing models. [V, VI, VII, VIII, and IX′] Non-compartmental Gompertz models for the groups α, β, γ, δ, and ε, respectively. [X, XVI, XVII, and XVIII] Compartmental Gompertz models for de protein, fat, water, and ash, respectively. [XI, XII, XIII, XIV, and XV] Compartmental Gompertz protein models for the groups α, β, γ, δ, and ε, respectively.

## Data Availability

Data will be made available on request.

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
