# Peer review of "Deciphering Seasonal Patterns in Animal Feeding: A Mechanistic Approach to Analyzing the Restricted Growth of Iberian Pigs"

_animals, 2024, doi:10.3390/ani14233431_

Round 1
Reviewer 1 Report
Comments and Suggestions for Authors
This study evaluated several mathematical models to predict growth patterns during the early growing and pre-finishing phases of Iberian pigs raised in semi-free-range environments. Overall, the methods are well-described and the results are well-present and well-discussed. I only have a few comments:
1, More details are needed for statistical analysis, such as the specific packages for model fitting in R.
2, The colorful photo (L253-255) seems unnecessary in a scientific paper.
3, The figure legends in page 12 should be under the figure, and the figures in page 13 lacks of figure legends.
• What is the main question addressed by the research? This study evaluated several mathematical models to predict growth patterns during the the early growing and pre-finishing phases of Iberian pigs raised in semi-free-range environments. • Do you consider the topic original or relevant to the field? Does it address a specific gap in the field? Please also explain why this is/ is not the case. Yes. This research will help to optimize the management practices in Iberian pig production systems • What does it add to the subject area compared with other published material? This research exhibited a good research by using mathematical models and big data analysis on pig production. • What specific improvements should the authors consider regarding the methodology? What further controls should be considered? More details should be added to the statistical analysis part as included in the review report submitted. • Are the conclusions consistent with the evidence and arguments presented and do they address the main question posed? Please also explain why this is/is not the case. Yes. Some novel mathematical models have been developed to describe the growth patterns of Iberian pigs. • Are the references appropriate? Yes. • Any additional comments on the tables and figures. Already included in the review report submitted.
Author Response
Thank you sou much for your time to the review of this paper and for your comments.
Coments 1: More details are needed for statistical analysis, such as the specific packages for model fitting in R.
Response 1: In the point 2.5, third paragraph has been explained the functions used to fit the models. In the Table S1 has been explained the fits of the Gompertz models in Rstudio.
Coments 2, The colorful photo (L253-255) seems unnecessary in a scientific paper.
Response 2. The photo could be unnecessary to understand the mathematical models. However, being an international article, this photograph could help to understand the management of the Iberian pig briefly explained in the introduction, as well as the need to carry out these researches precisely because of this peculiar management.
Coments 3. The figure legends in page 12 should be under the figure, and the figures in page 13 lacks of figure legends.
Response 3. Now it is OK, thank you for your comments.

Reviewer 2 Report
Comments and Suggestions for Authors
Dear Authors
It was interesting to read and review your manuscript titled, “Deciphering Seasonal Patterns in Animal Feeding: A Mechanistic Approach to Analysing the Restricted Growth of Iberian Pigs”.
This manuscript holds a scientific relevance towards sustainable swine production, improving pig welfare, enhancing product quality, and contributing to environmental conservation, particularly within the context of the Iberian pig's unique farming system. In addition, it will attract readership across different ongoing investigations in the field. However, I have a few suggestions:
L37 -38: For centuries, animal scientists have been trying to unravel the 'mysteries' of animal growth, where intra-species differences are mainly determined by genetics and nutrition. Mathematical models play a crucial role in understanding the main processes involved in 39 animal production [1], especially in predicting nutrient requirements [2], feed intake, and 40 animal growth [3].
Rewrite: For centuries, animal scientists have sought to uncover the complexities of animal growth, where variations within a species are largely influenced by genetics and nutrition. Mathematical models have been used to uncover the underlying processes involved in animal production [1], particularly in predicting nutrient requirements [2], feed intake, and growth [3].
L147 – 152: Define the parameters in each equation. e.g.
Where: y =??
(𝑡)=???
𝐴𝑒=???
𝑒𝑏=???
𝑘=??, and so on…
Kind regards.
Author Response
Thank you for your helpfull comments.
We have replaced our first paragraph with your suggestion. The authors had been discussing how to present this information. Thank you for improving the text.
We are defined all the parameters in the text.
Thanks again.
Reviewer 3 Report
Comments and Suggestions for Authors
Esquiliche et al. established a model that can be served as an effective tool for understanding, optimizing strategies, and enhancing animal growhth prediction. Overall, this is a good study on an important topic.
1.Please provide the analysis coding program and raw data in the supplemtal materials.
Author Response
Thank you for your review.
We have included the table S1 with the R codes used to create the models. We named in the "Data Availability Statement": "Data will be made available on request"
Thanks again
